

# Hydrogeological effects of dredging navigable canals through lagoon shallows. A case study in Venice

Pietro Teatini[1,2,3], Giovanni Isotton[2], Stefano Nardean[1], Massimiliano Ferronato[1,2], Annamaria Mazzia[1], Cristina Da Lio[3], Luca Zaggia[3], Debora Bellafiore[3], Massimo Zecchin[4], Luca Baradello[4], Francisco Cellone[5], Fabiana Corami[6], Andrea Gambaro[6,7], Giovanni Libralato[7,8], Elisa Morabito[6], Annamaria Volpi Ghirardini[7], Riccardo Broglia[9], Stefano Zaghi[9], Luigi Tosi[3]

[1]Department of Civil, Environmental and Architectural Engineering, University of Padova, via Trieste 63, 35121 Padova (PD), Italy
[2]M3E S.r.l., via Giambellino 7, 35129 Padova (PD), Italy
[3]Institute of Marine Sciences, National Research Council, Arsenale Tesa 104, Castello 2737/F, Venezia 30122, Italy
[4]National Institute of Oceanography and Experimental Geophysics (OGS), Borgo Grotta Gigante 42/c, 34010 Sgonico (TS), Italy
[5]Centro de Investigaciones Geologicas, Consejo Nacional de Investigaciones Científicas y Tecnicas, Diagonal 113 N°275, B1904DPK, La Plata, Argentina
[6]Institute of the Dynamics for the Environmental Processes, CNR-IDPA, Campus Scientifico -Ca' Foscari University of Venice, Via Torino, 155, 30172, Mestre-Venezia, Italy
[7]Department of Environmental Sciences, Informatics and Statistics, Ca' Foscari University of Venice, Via Torino, 155, 30172, Mestre-Venezia, Italy
[8]Department of Biology, University of Naples Federico II, Complesso Universitario di Monte S. Angelo, Via Cinthia ed. 7, 80126 Naples, Italy
[9]Institute of Italian Ship Model Basin, National Research Council, Via di Vallerano 139, 00128 Rome, Italy

*Correspondence to*: Pietro Teatini (pietro.teatini@unipd.it)

**Abstract.** For the first time a comprehensive investigation has been carried out to quantify the possible effects of dredging a navigable canal on the hydrogeological system underlying a coastal lagoon. The study is focused on the Venice Lagoon, Italy, where the Port Authority is planning to open, a new 10-m deep and 3-km long canal to connect the city passenger terminal to the central lagoon inlet thus avoiding the passage of large cruise ships through the historic centre of Venice. A modelling study has been developed to evaluate the short (minutes), medium (months), and long (decades) term processes of water and pollutant exchange between the shallow aquifer system and the lagoon, possibly enhanced by the canal excavation, and ship-wakes. An in-depth characterization of the lagoon subsurface along the channel has supported the numerical modelling. Piezometer and sea level records, geophysical acquisitions, laboratory analyses on groundwater and sediment samples (chemical analyses and ecotoxicity testing), and the outcome of 3D hydrodynamic and computational fluid dynamic (CFD) models have been used to set-up and calibrate the subsurface multi-model approach. The numerical outcomes allow to quantify the groundwater volume and estimate the mass of anthropogenic contaminants (As, Cd, Cu, Cr, Hg, Pb, Se) likely leaked from the nearby industrial area over the past decades, and released into the lagoon from the canal bed by the action of depression waves generated by ships. Moreover, the model outcomes help to understand the effect of the





hydrogeological layering on the propagation of the tidal fluctuation and salt concentration into the shallow brackish aquifers underlying the lagoon bottom.

# 1 Introduction

Coastal lagoons are transient ecosystems highly sensitive to changes in sedimentation, sea level rise, and land subsidence. In many cases, their evolution over the last centuries has been strongly impacted by human activities. The use of these peculiar ecosystems, for fish and shell farming, tourism, transportation of people and goods, has usually contrasted with the preservation and protection of habitat and biodiversity (Kennish et al., 2010). One typical intervention in coastal systems is dredging of canals and inlets, which may be performed to increase the water volume exchanged with the sea (Gong et al., 2008) or for navigation purposes (Fortunato and Oliveiram, 2007; Healy et al., 1996). Elsewhere, dredging has been used as a source of fill material for adjacent upland development and land reclamation (López et al., 2013).

The opening of waterways in shallow coastal waterbodies and lagoons have facilitated navigation for centuries providing sheltered routes and permitting safe access to inland ports and harbours. However, the progressively increasing tonnage of vessels and sediment dynamics requires port authorities worldwide to implement dredging programs to guarantee navigability and to open new shipping channels to allow larger traffic.

In many cases, this practice has led to environmental deterioration, by changing the flushing efficiency of the canal system, aggravating salinity stratification, re-suspending fine sediments, pollutants and nutrients, which are responsible for eutrophication, hypoxic events, and increasing contamination and release of pollutants (e.g., Newell et al., 1998). For example, the combined impacts of increased turbidity and physical removal or burial during dredging, caused the loss of approximately 81% of the seagrasses in Tampa Bay, Florida (Erftemeijer and Lewis, 2006). Moreover, canal dredging has been responsible for significant hydro-morphological impacts in coastal lagoons. A significant example is the case of Aveiro Lagoon in Portugal, where two centuries of channelization, jetty breakwater construction, and dredging have led to a progressive shift from the original fluvially dominated system into the present tidally dominated one. The associated stresses imposed by increased tidal currents have led to important changes in the sedimentary regime (Duck and da Silva, 2012).

In the Venice Lagoon, Italy, engineers and administrations have planned dredging works for centuries, creating a series of canals for navigation and reclaimed land for urban expansion and industrial settlement (Balletti 2006; D'Alpaos, 2010). The last major navigable canal, the Malamocco-Marghera Industrial Canal (MMIC), was excavated in 1970 to connect the Porto Marghera Industrial Zone (PMIZ) on the mainland with the Adriatic Sea through the Malamocco inlet (Fig. 1a). A large number of studies developed over the last decade has demonstrated that the MMIC and the navigation of large vessels through the lagoon shallows have likely been main causes for the morphological deterioration observed in the central lagoon as deepening of the tidal flats, marshland erosion, and sediment loss (e.g., Amos et al., 2010; Carniello et al., 2009; Ferrarin et al., 2013; Marani et al., 2011; Molinaroli et al., 2009; Tambroni and Seminara, 2006). The Venice Port Authority has recently planned the excavation of a new approximately 3-km long and 10-m deep navigation canal (called Marghera-Venice





Canal, MVC, in the sequel) to reroute vessels along the MMIC and reach the passenger terminal located in the southwestern part of the historic center (Fig. 1b). The intervention should avoid the transit of large cruise liners though the historic center of Venice. At present, more than 500 cruise ships enter the lagoon each year (http://www.vtp.it/en/company/statistics/) and this traffic will add to the already intense commercial traffic that is based on the MMIC.

Despite the large research effort dedicated to the understanding of the fresh- groundwater exchange in coastal aquifers (e.g., Li et al., 1999; Michael et al., 2005; Nakada et al., 2011; Qu et al., 2014), studies developed in the past have never addressed the evaluation of possible effects of excavating navigable canals through tidal flats on the underlying hydrogeological system. However, in-depth investigations using direct measurements (isotopes, benthic chambers), geophysical surveys, and modelling simulations revealed that submarine groundwater discharge (SGD) may provide

considerable fresh-water inputs to coastal waterbodies (e.g., Rapaglia et al., 2010; Wang et al., 2015) and may be the primary pathway for nutrients and other contaminants to enter coastal lagoons (e.g., Rapaglia, 2005; Rocha et al., 2016; Santos et al., 2008; Tait et al., 2013).

The primary objectives of this study are to investigate how the construction of a new large navigable canal through tidal flats affects i) the groundwater flow and quality of the shallow aquifers underlying the lagoon bottom and ii) the exchange of

water and chemicals from the subsurface to the surface waterbodies. Our research focuses on the Venice Lagoon as a representative case study. Based on the available knowledge on the surface and subsurface lagoon environment, the following issues had to be considered in the context of the study:

–   the quality of the surficial water, mainly its salinity, with respect to the groundwater. Can an eventual cut of impervious layers enhance saltwater leakage beneath the lagoon bottom?

–   the presence of chemicals in the groundwater below the lagoon bottom and the sediment toxicity due to leakage from the industrial and urban centres located in the lagoon surroundings. Are contaminants present also along the MVC designed path? Which is their mobility? And, can the MVC excavation determine their release into the lagoon, also favoured by SGD from exposed sub-surficial heterogeneities?

–   the evolution of water level in the lagoon canals and flats due to the transit of large vessels. How do the solitary

waves associated to the passage of large vessels in the navigation channel influence the flow and contaminant transfer between the subsurface and surficial systems?

In this study, for the first time, we explore in detail these issues, improving the understanding of the interaction between the subsurface and surface waters in coastal systems, and providing quantitative evaluations for the specific case study. This is carried out through an accurate investigation on of the lagoon environment along the MVC trace and the use of uncoupled

and coupled density-dependent groundwater flow and transport simulators.

The paper is organized as follows. Section 2 presents the main results obtained to characterize the subsurface system of the Venice Lagoon along the MVC and the factors forcing its dynamics. The numerical approach used to perform the hydrogeological modelling study is revised in Section 3 together with the description of the model set-up. The results obtained by the computations are presented in Section 4, pointing out the effect of the canal dredging by comparing the





model solutions in the present (i.e., without the canal) and in the planned (i.e., with the canal) scenarios. A conclusive section (section 5) discusses the main outcomes and draws the principal conclusions of the study.

## 2 Hydro-stratigraphic, chemical, ecotoxicological, and hydrodynamic setting of the central Lagoon of Venice

### 2.1 Hydro-stratigraphic characterization

About 40 km of very-high-resolution seismic (VHRS) lines (Fig. 1b) were collected by a boomer system equipped with an electro-dynamic plate and a single-channel streamer. The latter consisted of 8 equidistant piezoelectric elements housed in an oil-filled tube and connected in series with a 2.8 m active array section (Tosi et al., 2009). The frequency bandwidth produced by the plate ranged from 0.4 to 9 kHz, thus allowing a decimetre resolution. Suitable floaters kept the streamer as shallow as possible to avoid destructive interference between reflected signals and multiple events from the air/water

interface. Because the investigated area is characterized by shallow water (< 1.5 m) and the conventional acquisition geometry (streamer towed behind a source) generates poor results using a single channel streamer, a transverse geometry was applied to collect more coherent events (Baradello and Carcione, 2008). The seismic data were processed by a conventional sequence, including initially a spherical divergence removing, secondly a time-variant gain, and finally a time-variant band-pass filter. The marine and boat waves degraded the reflection signal in a number of profiles. This effect was

mitigated by computing a mean trace in a given interval, cross correlating it with the single traces, and applying the corresponding time shift as a static correction.

The interpretation of the seismic units with the support of stratigraphic data obtained through ten continuous 10 m-long cores (Fig. 1b) specifically drilled for the study allowed to sketch the hydro-stratigraphic setting of the lagoon subsoil along the MVC designed trace. In addition, a number of geophysical, lithological, sedimentological, and geotechnical information

available from previous investigations (Fabbri et al., 2013; Madricardo and Donnici, 2014; Teatini et al., 2011; Tosi et al., 2007, 2011; Zecchin et al., 2011, 2014) was reprocessed to characterize the architecture of the deeper deposits (down to a depth of approximately 50 m below msl) and contextualize the investigated area within a regional hydrogeological framework. Fig. 2 shows two interpreted VHRS lines, namely Section-1 and Section-2, orthogonal to the MVC trace. The seismic survey revealed a high heterogeneity of the lagoon subsoil due to a number of buried paleo-channels, whose

presence would not have been highlighted only by the core samples. This study is focused on these two sections, each of them crossing a 10-m depth borehole. The borehole coordinates are the following: 286'467 East 5'034'726 North (Section-1) and 287'843 East, 5'035'184 North (Section-2), UTM33 WGS84.

Combining the new hydro-stratigraphic information and those available from previous studies, it has been possible to characterize the hydro-stratigraphic system and identify three main permeable units down to about 50 m depth beneath the

lagoon bottom (Fig. 2). In the easternmost Section-1, the shallowest sandy unit (Aqf-1) is 7 to 10 m thick and lies below a few meter thick silty-muddy layer (Aqt-1). Aqf-1 is almost continuous in the central and southern parts of the investigated area and reduces northward where silty-clay deposits prevail. This aquifer represents a buried paleo-channel, whose direction



is from the industrial zone to the historical centre of Venice. The top of Aqf-1 represents the Holocene–Pleistocene limit. This is marked by an erosional unconformity generally made by a metric cemented clayey layer (Tosi et al., 2007), known in the Venice area as ''caranto''. A second sandy unit (Aqf-2) is generally confined below a 2-3 m-thick silty-clayey layer (Aqt-2). In the central-western part of the study area where Section-2 is located, Aqt-2 lacks because of paleo-channel

incisions, and Aqf-1 and Aqf-2 are undifferentiated. At the bottom of Aqf-2, a second quite continuous aquitard (Aqt-3) confines a third aquifer (Aqf-3), which has a regional extent. The Aqf-3 depth ranges between -25 and -35 m below msl and the thickness amounts to about 8 m.

## 2.2 Hydrogeological characterization

Despite the large effort carried out in the past to define the hydrogeological setting of the shallow lagoon subsurface, scarce

information is available in the study area because the characterization was mainly concentrated along the littoral strips. A 100-km long airborne electromagnetics (AEM) survey carried out in 2009 between Venice and the industrial area pointed out the important hydrogeologic function played by the caranto (Teatini et al., 2011). The AEM investigation provided resistivity information from the lagoon bottom down to about 120–140 m depth and clearly showed that the caranto reduces or precludes the downward leakage of seawaters. Groundwater with a salt concentration comparable with the marine waters

(resistivity $\rho$ ranging between 0.1 and 1 $\Omega \cdot$m) is encountered below this impermeable layer only where it is missing because of natural erosion or canal dig. The measured $\rho$ values increased to 2-10 $\Omega \cdot$m in the shallowest portion of the Pleistocene deposits, with almost freshwater groundwater ($\rho$>10 $\Omega \cdot$m) below 10 to 30 m depth depending on the position. This strategic role may have considerable implications for the MVC project.

Groundwater and hydrogeologic properties in the upper 10 m depth have been investigated by a Casagrande piezometer

installed at the bottom of the boreholes along Section-1 and Section-2. Each monitoring station was instrumented by two CTD-Divers, one placed within the borehole and connected to the Casagrande cell and the other fixed outside the borehole casing at the lagoon bottom. The configuration allowed the simultaneous monitoring of electrical conductivity (EC), temperature (T), and pressure (P) in Aqf-1 and lagoon waters, thus revealing the possible relationship between the surface and subsurface waterbodies.

Fig. 3 shows an example of the recorded Aqf-1 pressure head and lagoon level during a few days in March 2016 at the Section-2 station. As expected, the fluctuation of the groundwater level is phased on the semi-diurnal tidal regime, with a gentle (10-15%) reduction of the wave height and a delay of 10 to 20 minutes on the maximum/minimum occurrence. Similar values were obtained at the Section-1 borehole. Concerning EC, the records are characterized by a negligible variability in time. With reference to a temperature T=25°C, EC=46-51 mS/cm and 47-53 mS/cm in the lagoon water at

Section-1 and Section-2, respectively. In Aqf-1, EC is approximately 15-20% higher in Section-1 and smaller Section-2, respectively, than the surface water. The differences suggest a possible groundwater contamination of anthropogenic origin in Section-1, which is close to PMIZ and the Tresse Islands made of muds dredged from the PMIZ canals, and the effect exerted by the caranto in reducing the aquifer salinization in Section-2.



## 2.3 Chemical characterization

Among the various pollutants, trace elements are of particular concern since they are up-taken by biota and may have toxic effects. Some trace elements are in the priority list and regulated by European directives, e.g. the 2000/60 EC (Water Framework Directive), and their national transpositions, e.g. the Ministerial Decree 260/2010 in Italy. Furthermore, laws and
decrees regulate the presence of trace elements in specific relation to dredging. Before any dredging, the chemical characterization together with an eco-toxicological evaluation must be carried out. Although in the past much interest has been focussed on total concentration of trace elements, it has been more recently accepted that assessing the mobility, the bioavailability, the bioaccessibility, and the toxicity of metals is fundamental (Schintu et al., 2016; Zhang et al., 2017).

Within this study, both the total concentration of trace elements and the geo-speciation, defined according to Ure et al.
(1993), were carries out on samples collected from the two reference boreholes along Section-1 and Section-2 (Fig. 1b). The geo-speciation was performed via the sequential extraction procedure (SEP) proposed by Tessier et al. (1979) and harmonized by Corami et al. (2009). SEP allows an operational classification of metals into four geochemical fractions with different mobility, bioavailability, and bioaccessibility. The labile fraction is characterized by the highest mobility and bioaccessibility, the residual fraction by the lowest mobility.

A detailed description of the preparative and analytical methods employed to assess the total concentration and to study the geo-speciation is beside the aim of this paper and the reader can refer to DAIS (2016). Shortly, a hydraulic corer was used to collect the samples. Dried aliquots of sediments, previously homogenized for each meter interval, were assessed for the total concentration of twelve trace elements (As, Be, Cd, Cr, Cu, Hg, Ni, Pb, Sb, Se, V, Zn). Unaltered aliquots were analysed by the harmonized SEP to quantify the labile and bioaccessible fraction of some trace elements, namely As, Cd, Cu, Cr, Hg, Pb,
and Se. Table 1 provides an example of the results obtained by the chemical characterization in terms of total and labile Cr concentrations versus depth measured in the Section-1 borehole. As a general feature, the labile and bioaccessible fraction increases with depth. This trend has been confirmed at Section-1for every trace element considered by this study and, to a lesser extent (Cd and Pb), in Section-2.

## 2.4 Ecotoxicological characterization

Ecotoxicity is of great interest in sediment assessment and management providing an integrated response related to the bioavailable and bioaccessible fraction of contaminants within the checked matrix (whole sediment, pore water and elutriate). The sediment samples collected in Section-1 and Section-2 were homogenised and sieved at 2 mm (in a $N_2$ atmosphere for pore water (PW) production). Ecotoxicity was investigated on sediments and aqueous extracts such as elutriates (E) and PWs prepared according to Arizzi Novelli et al. (2006) and Losso et al. (2009), respectively. A battery of
toxicity tests was used including acute (A) and (sub-)chronic ((S)C) endpoints with *Vibrio fischeri* (A) (ISO 21338:2010, whole sediment) and *Corophium orientale* (A) (ISO 16712:2005, whole sediment), *Phaeodactylum tricornutum* (C) (ISO 10253:2006, growth inhibition test on E and PW), *Crassostrea gigas* (SC) (ISO 17244:2015, embryotoxicity test on E and





PW), *Mytilus galloprovincialis* (SC) (ISO 17244:2015, embryotoxicity test on E), and *Paracentrotus lividus* (A) (Volpi Ghirardini et al. (2005), sperm-cell toxicity on PW and embryotoxicity on E). Possible confounding factors, like $NH_3$ and $S_2$, were assessed as well (Libralato et al., 2008). Toxicity data were managed according to their relative standard protocol and integrated in a final judgement considering the worst-case scenario approach according to the precautionary principle.

The results evidenced a toxicity range from absent/low (acute tests) to very high (i.e. all embryotoxicity tests) considering as ranking tools the toxicity scales set up on a species-by-species basis for Venice Lagoon sediments (Losso et al., 2010). According to the worst-case scenario approach, sediment presented very high levels of toxicity independently from the core depth (from 0 to -8 m below the lagoon bottom) evidencing potential high risk for early life stages of sensitive marine organisms like bivalves and sea urchins. The increase of labile and bioaccessible fractions of metals along depth gradients
could explain this evidence being toxicity the response to the real bioaccessibility of contaminants in aqueous media.

**2.5 Hydrodynamic characterization**

Apart from the natural tidal regime, a certain effect on the hydrogeological system in the surrounding of deep channels is expected to be driven by long inverse solitary waves associated to the passage of large vessels in the navigation channel and known as depression wakes, or Bernoulli wakes (Rapaglia et al., 2015). Ship-wakes were characterized by means of water
level measurements made with pressure sensors and turbidity meters deployed along a profile on the channel side and the surrounding mudflat together with a modelling chain capable of reproducing the hydrodynamic patterns in the channel around the hull of the moving ship, and the propagation of the depression wake on the tidal flat.

Tidal level and ship-induced depression wakes as well as short-period boat wakes were measured with a pressure sensor with logger (Solo D/Wave, RBR, Canada) immersed at a depth of approximately 4 m on the eastern side of the MMIC. Pressure
was recorded by the instrument at a sampling frequency of 16 Hz and converted in depth data. The experimental setup also included an electromagnetic current meter deployed at the bottom of the navigation channel which recorded water level, current speed and direction at an acquisition frequency of 2 Hz. Simultaneously, an automatic identification system (AIS) receiver permitted to acquire traffic data for the area, relating every observed event to the specific ship in transit in the measurement section. Tidal levels were referred to the local datum, while depression wakes were calculated as the difference
between maximum and minimum levels at the passage of a ship. Fig. 4a shows the ship-wake generated by the passage of a commercial vessel (Fig. 4b) on April 6, 2016. A relatively small rise of the water level (~0.1 m) before the ship reaches the measurement section is followed soon after the transit by a significant depression (~1.6 m) that develops within the channel and propagates in the mudflat. The depression wake lasted for about 80 s.

The simulation of the pressure and velocity fields around the hull of the ship was carried out using the uRaNSe-Xnavis
simulator (Broglia et al., 2014; Di Mascio et al., 2007, 2009). It is a finite volume solver based on the discretization of free surface, incompressible, viscous, high-Reynolds-number fluid equations (unsteady Reynolds averaged Navier-Stokes equations). The fluid-dynamical field is discretized using the overlapping grid approach, with an increase in spatial resolution close to the hull and the free surface, then degrading in the channel close to the tidal flat. The channel and the





lateral zone where the bathymetry is deeper than 2 m represent the uRaNSe-Xnavis domain. An example of the computational grid is shown in Fig. 5a. The model was tested and calibrated using the data recorded along the MMIC and then applied to forecast the movement of a typical cruise along the planned MVC.

The high resolution CFD steady state dimensionless results, in terms of water level and velocity, were interpolated on a regular $0.7 \times 0.7 \times 0.4$ m grid, dimensionalized, and used to force a shallow water hydrodynamic code called SHYFEM (Shallow Water Hydrodynamic Finite Element Model) (Bellafiore and Umgiesser, 2010; Umgiesser et al., 2004). The finite element grid representing the portion of the Venice Lagoon with the MVC trace is shown in Fig. 5b. At each time step the uRaNSe-Xnavis results is geolocalized within the SHYFEM finite element grid and moved along the ship trajectory with a defined speed $s$. Water levels and 3D velocities force the system. SHYFEM solves the shallow water equations in the whole computational basin shown in Fig. 5b except the area covered by the uRaNSe-Xnavis results, which is the box including the ship and a portion of the surrounding channel (Fig. 5a), where a combined technique imposing and nudging input data is adopted. The evolution of the water level for the whole period covered by the passage of the ship and in the whole modelled basin encompassing both the MVC and the tidal flat is produced. Time series of water level along Section-1 and Section-2 were extracted and used as forcing in the hydrogeological models.

A set of four scenarios were produced, considering a typical liner ship having the geometrical characteristics provided in Table 2 and moving in the system with $s = 3.1, 4.9, 5.9, 7.7$ knots. These velocity values cover a range from very low speeds, which are typical of in port operations and manoeuvring, to speeds above the navigation limit (6 knots) typically measured in the channel and recorded in our AIS logs. The water level depression produced by the passage of the ship has a wide range of amplitudes, depending on $s$ and shows different behaviours approaching the tidal flat. As expected, the ship passage produces a depression whose maximum value increases with $s$ and decreases with the distance from the canal center (Table 3 and Fig. 6). Notice that, a cruise ship generates a depression wake that is almost half of that produced by the cargo shown in Fig. 4 when a comparable speed (i.e. 7.7 versus 8.1 knots) is assumed. The lateral depression is weakly (less than 20%) attenuated within 100 m from the channel, in the close tidal flat, for the two highest speed values. At a distance of about 600 m from the channel, the height of the depression in the mudflat remains significant only for the $s = 7.7$ knots case. For lower ship velocities, the residual signal over the mudflat is quite small.

## 3 Modelling approach and set-up

Short (from minutes to hours) and medium (i.e., months) term simulations addressing the effects of the tidal fluctuations and ship-wakes have been carried out by a flow and transport uncoupled approach using the subsurface modules FLOW3D and TRAN3D of the finite element CATchment HYdrology Flow-Transport (CATHY_FT) model (Camporese et al., 2010; Weill et al., 2011). The mixed hybrid finite element-finite volume COUPHYB simulator (Mazzia and Putti, 2006) for the solution of density-dependent flow and transport has been used to perform long-time (i.e., decades) analyses of seawater leakage into the aquifer system below the lagoon bottom.



The numerical simulations were carried out on 2D vertical sections, in particular along Section-1 and Section-2 (Fig. 1b) that can be considered representative of the lagoon hydrogeological setting with respect to the MVC excavation. Indeed, the MVC bottom planned at -10.5 m below msl is bounded by an impermeable unit in Section-1 and is located in the middle of the sandy Aqf-1/2 in Section-2 (Fig. 7a). The effects of MVC was investigated by comparing the model results in the present (i.e., without the canal) and in the planned (i.e., with the canal) conditions. The model domains extend 600 m in the horizontal direction, for 300 m on each side of the MVC axis, and vertically from the lagoon bottom down to -50 m above msl. Fig. 7b shows the triangular grid used to discretize Section-1. The distribution of the geologic layers as reconstructed in Fig. 2 was accurately reproduced within the model. The element size ranges from 0.5 m in the central zone around the MVC trace to 5.0 m in the outer portions. The nodes totalled about 18'000 and 70'000 in the present state and with the MVC, respectively, with approximately 35'000 and 140'000 triangles.

### 3.1 Tidal and ship pressure fluctuations

The evolution of the pressure and velocity fields in the shallow subsurface due to water level fluctuations in the lagoon was simulated by FLOW3D neglecting the possible effects of different groundwater and surface water salinity. Indeed, density-driven processes are characterized by a much longer characteristic time than those typical of tidal regime and ship-wakes. FLOW3D solves the groundwater flow equation in saturated conditions:

$$\vec{\nabla} \cdot \left[ K\, \vec{\nabla} h \right] = S_s\, \frac{\partial h}{\partial t} + q \,, \tag{1}$$

where $K$ and $S_s$ are the saturated hydraulic conductivity [LT$^{-1}$] and the specific elastic storage [L$^{-1}$] of the porous medium, respectively, $h = z + p/(\rho_0 g) = z + \psi$ is the hydraulic head [L], $z$ the vertical coordinate directed upward [L], $p$ the pressure [ML$^{-1}$T$^{-2}$], $\rho_0$ the fresh water density [ML$^{-3}$], $g$ the gravitational acceleration [LT$^{-2}$], $t$ the time [T], $q$ a source or sink term [L$^3$L$^{-3}$T$^{-1}$], and $\vec{\nabla}$ the gradient operator [L$^{-1}$]. In FLOW3D Eq. (1) is solved using linear Galerkin finite elements (FE), with triangular elements and a weighted finite difference time integration scheme (Paniconi and Putti, 1994).

FLOW3D was initially used to calibrate the hydrogeologic properties of the upper units. This was carried out by running the model in the present configuration and matching the pressure records available at the two 10-m deep piezometers. Dirichlet conditions representing the observed tidal regime over spring 2016 were imposed on the top boundary, which constitutes the lagoon bottom, a constant head $h = 0.5$ m above msl, i.e. the average water level over the monitoring period, is prescribed on the lateral bounds, and zero flux through the bottom. The $K$ and $S_s$ obtained by the calibration for Aqt-1 and Aqf-1 are presented in Table 4. They are in good agreement with values obtained in previous modelling studies carried out in the Venice area (Castelletto et al., 2015; Paris et al., 2011) and hence used to characterize also the deeper layers.

The calibrated model was then used to quantify the effect of the MVC in term of pressure and flow field on the subsurface in relation to:





- tidal regime: the same boundary conditions used for the model calibration were applied on the domain with the MVC and the results obtained with the two configurations were compared. The simulations were carried out using a time step $\Delta t = 360$ s;

- ship-wakes: we investigate the effects induced by both the commercial vessel of Fig. 4, which represents an extreme of the perturbations possibly stressing the system, and a cruise vessel with $s = 7.7$ knots. The initial conditions were derived from the FLOW3D outcome obtained in the previous simulations investigating the tidal regime. The boundary conditions are represented by the measured (Fig. 4a) and simulated (Fig. 6) water levels for the nodes corresponding to the MVC bottom and slopes. The results of the SHYFEM model at a distance of 300 m from the MVC were used to fix the behaviour of the water level for the nodes on the lateral boundaries. A linear interpolation, both in space and in time, between the canal and the 300-m aside values was used to derive the water level on the nodes located on the tidal flats. Similar to the previous case, the bottom boundary was assumed impermeable. The ships transit after 2700 s (approximately 45 minutes) from the inception of the simulation, which spans a total time equal to 3000 s using $\Delta t = 1$ s.

### 3.2 Contaminant transport

Transport processes in the subsurface of non-reactive chemicals are described by the classical advection–dispersion equation (Bredehoeft and Pinder, 1973):

$$\vec{\nabla} \cdot \left[ D \vec{\nabla} c - \vec{v} c \right] = \phi \frac{\partial c}{\partial t} - q_c , \tag{2}$$

where $D$ (Bear, 1979) is the dispersion tensor accounting for both mechanical dispersion and molecular diffusion [$L^2 T^{-1}$], $c$ the subsurface solute concentration [$ML^{-3}$], $\vec{v}$ the Darcy velocity vector [$LT^{-1}$], $\phi$ the porosity [-], and $q_c$ a term incorporating an external solute sink or source [$ML^{-3}T^{-1}$].

Similarly to FLOW3D, TRAN3D solves Eq. (2) using a Galerkin FE approach and a weighted finite difference time integration scheme (Gallo et al., 1996). TRAN3D was used to quantify the possible exchange of the contaminants detected in the subsurface between the groundwater and the surface waters along the MVC trace. In particular, the attention was focused at the ship-wakes that, being strongly asymmetric, favour the contaminant outflow into the lagoon waters much more than the tide fluctuations, which are almost symmetric with respect to the mean sea level. The velocity field at each time step is provided by the outcome of FLOW3D. Based on the general outcome of the chemical characterization, the following simplifying assumptions were adopted in the modelling set-up:

- $c$ represents the concentration of the sole labile and bioaccessible fraction, i.e. the contaminant portion that can be reasonably assumed in equilibrium with the concentration in the groundwater;

- the $c$ values measured at the two wellbores are representative of the initial contaminant distribution within the whole modelling domains;





- Table 5 provides the $c$ values averaged between the depth range from -10 to -5 m below msl. These concentrations were assumed as uniformly distributed within Aqf-1 (Section-1) and Aqf-1/2 (Section-2). Based on the available data, in Aqt-1 $c$ can be grossly assumed to be equal to 20% of the value in the underlying aquifer. This $c$ distribution is used as initial condition in the transient transport simulations;

- $c = 0$ below Aqf-1 in Section-1 and Aqf-1/2 in Section-2, i.e. Aqt-2 precludes the anthropogenic contamination at larger depth.

The simulations were carried out by normalizing the actual $c$ values with respect to the initial concentrations in the shallower aquifer. This allowed running the transport model independently from the specific contaminant, with the quantification of the actual concentration in the subsoils and mass expelled into the lagoon carried out a-posteriori for each species multiplying

the modelling outcome by the values provided in Table 5. A value $c = 0$ has been imposed on the top boundary, i.e. the concentration of the metals in the lagoon water has been assumed negligible, with $\partial c / \partial n = 0$ along the lateral and the bottom boundaries. Due to the lack of specific information, a proper sensitivity analysis was carried out on the longitudinal ( $\alpha_L$ ) and transversal ( $\alpha_T$ ) dispersivity [L], which defines the dispersion tensor $D$ (Bear, 1979). In particular, a value of $\alpha_L$ ranging from 0.1 m to 10 m has been evaluated, with the ratio $\alpha_L / \alpha_T$ varying from 0.1 to 1. Table 6 summarized the

investigated scenarios.

The transport process was investigated over a multiple ship passage. Specifically, the transit of $N_v = 1000$ vessels was simulated by concatenating the relative FLOW3D outcome 1000 times. According to the information available from the Venice Port Authority and summarized in the introduction, this simulation should represent a period of approximately 1 year.

**3.3 Density-dependent interactions**

As reported above, the available hydrogeological investigations revealed that the subsurface of the Venice Lagoon is characterized by significant stratification in term of water salinity, reflecting the layering of the sedimentary sequence. Dredging of new canals in such an environment can cut impermeable units, producing a certain mixing of the salt concentration between the shallowest contaminated units and the fresher underlying layers. Due to the lack of significant

pressure gradient between the various geologic layers, the difference in groundwater density is likely the main driver of salt transport.

Following Bear (1979), the mathematical model of density-dependent flow in aquifer systems can be written using the equivalent freshwater pressure head $\psi$ and the salt concentration normalized with respect to its maximum value $c$. Defining the density $\rho$ and the dynamic viscosity $\mu$ of the saltwater through the reference density $\rho_0$ and dynamic viscosity $\mu_0$:

$$\rho = \rho_0\left(1 + \varepsilon c\right), \tag{3}$$

$$\mu = \mu_0\left(1 + \varepsilon' c\right), \tag{4}$$





where $\varepsilon = (\rho_s - \rho_0)/\rho_0$ and $\varepsilon' = (\mu_s - \mu_0)/\mu_0$, with $\rho_s$ and $\mu_s$ the density and viscosity of the solution at $c = 1$, respectively, the equations of mass conservation for the coupled flow and transport model in porous media can be written as:

$$\vec{\nabla} \cdot \left[ K \frac{1+\varepsilon c}{1+\varepsilon' c} \left( \vec{\nabla}\psi + (1+\varepsilon c)\eta_z \right) \right] = \sigma \frac{\partial \psi}{\partial t} + \phi\varepsilon \frac{\partial c}{\partial t} + \frac{\rho}{\rho_0} q* + q$$
$$\vec{v} = -K \frac{1+\varepsilon c}{1+\varepsilon' c} \left[ \vec{\nabla}\psi + (1+\varepsilon c)\eta_z \right] \qquad , \qquad (5)$$
$$\vec{\nabla} \cdot \left[ D\vec{\nabla}c - \vec{v}c \right] = \phi \frac{\partial c}{\partial t} - qc^*$$

In Eq. (5), $\sigma = S_s(1+\varepsilon c)$ is the general storage term, the vector $\eta_z$ is equal to 0 along the $x$ and $y$ directions and to 1 along

the $z$ direction, $q*$ is the injected and $q$ the withdrawn volumetric flow rate, and $c*$ is the normalized concentration of salt in

the injected/extracted fluid.

In COUPHYB simulator (Mazzia and Putti, 2006), the system (5) is solved numerically using a mixed hybrid finite element

scheme for the flow equation and a mixed hybrid finite element-finite volume time-splitting-based scheme for the transport

equation. This approach is computationally effective and accurate, introducing minimal numerical diffusion even in the

absence of physical dispersion, and when the process is advection dominated or density changes yield instabilities in the

flow field. COUPHYB was applied to the same triangulation shown in Fig. 7b, with the solutions in terms of $\psi$ and $c$

provided on the mesh elements and in term of $\vec{v}$ on the element faces, which amount to about 60'000 and 200'000 in the

present state and with the MVC, respectively.

Based on the hydrogeological information, the simulations were carried out starting from an initial $c$ distribution where the

first buried clay layer (Aqt-2 in Section-1 and Aqt-3 in Section-2) prevented the downward propagation of the saltier water.

Therefore, $c = 0$ and $c = 0.7$ below and above the top of the sealing layer, respectively. The latter value was obtained based

on the data published in Teatini et al. (2011). Concerning the boundary conditions, $\partial c/\partial n = 0$ was prescribed along the

lateral and the bottom boundaries, with $c = 1$, i.e. the seawater concentration, imposed on the nodes corresponding to the

lagoon and MVC bottom. For the flow equation, the bottom was considered impermeable and a constant tidal level equal to

0.5 m above msl is assumed in the lagoon. The $\psi$ values on the top and lateral boundaries were computed in agreement with

the $c$ initial distribution and using $\rho_s/\rho_0 = 1.035$ ($\varepsilon = 0.035$). Moreover, $\varepsilon' = 0.231$, $\alpha_L = 1.0$ m and $\alpha_T = 0.1$ m were

assumed. The simulations covered a time interval equal to 10 years.



## 4 Results

### 4.1 Tidal effects

The dredging of a new relatively deep channel in a tidal environment can perturb the natural pressure and flow fields in the shallow subsurface. Quantification is obtained by comparing the results provided by the calibrated FLOW3D for the two simulated sections in the present condition and after the MVC excavation. Fig. 8 shows the behaviour of the pressure at a depth of 13 m below msl in correspondence of the MVC symmetry axis. The point is located within Aqt-2 and Aqf-1/2 in Section-1 and Section-2, respectively. The effect of the different stratigraphic sequence is obvious, with an approximately 85% reduction of the pressure fluctuation within the clayey layer with respect to the oscillation of the lagoon level (Fig. 8a). The MVC dig reduces the time lag and the attenuation of the perturbation at depth. These effects are quantitatively negligible and develop only in the surroundings of the channel (Fig. 9).

### 4.2 Ship-wake effects

Although a depression wave caused by a ship transit develops over a period of a couple of minutes (Figs. 4a and 6), which usually is a very short time for hydrogeological processes, its height is sufficient to affect significantly the groundwater pressure and flow fields in the proximity of the channel bottom.

Fig. 10 provides the pressure distribution in the surrounding of the MVC, Section-2, computed by FLOW3D at a few significant time steps during the transit of Cargo-Hazard A (Fig. 4b). The pictures clearly shows how the pressure gently rises before the ship passage, significantly decreases soon after the transit, and then recovers with a gradient that change its sign during each phase. The pressure change affects the portion of the subsoil down to the top of the first clay layer below the channel bottom, and extends laterally up to about 30 m from the channel slope (Figs. 11a and 11b for Section-1 and Section-2, respectively). The typical short duration of such events precludes the propagation of the pressure change far from the channel edges. The velocity field in correspondence of the maximum depression is provided in Figs. 11c and 11d. The ship generates a sort of "piston effect" with an efflux distributed along the whole channel bottom and slope. The maximum values of the velocity amount to $1.1 \times 10^{-5}$ m/s and $0.6 \times 10^{-5}$ m/s for Section-1 and Section-2, respectively. The highest velocities are computed in Section-1 due to the vicinity of the Aqt-2 top to the MVC bottom and the consequent pressure gradient much larger than in Section-2.

The seepage from the MVC bottom and slope (between times T2 and T4 in Fig. 10) amounts to $3.25 \times 10^{-2}$ m$^3$ and $2.63 \times 10^{-2}$ m$^3$ per meter length of the channel for Section-1 and Section-2, respectively. Considering the total length of approximately 3 km, each commercial vessel produces a cumulative groundwater volume flowing into the MVC totalling ~100 m$^3$ or ~80 m$^3$ assuming Section-1 or Section-2, respectively, as representative of the lagoon subsurface. Therefore, a value of about 90 m$^3$ can be estimated on the average.

The same computation was carried out for the ship-wake caused by a cruise ship moving at 7.7 knots. The results provided by the hydrodynamic model (Fig. 6) were used to force FLOW3D. The computed subsurface pressure and velocity fields are





qualitatively similar to those previously described (Figs. 10 and 11), with smaller values determined by the lower wave height used as forcing factor. The average efflux along the whole MVC reduces to ~45 m$^3$ per ship transit.

The effect of the transit of several ships along the MVC in term of contaminant release from the subsurface into the lagoon was investigated thorough TRAN3D. The behaviour of the released mass $m^*$ versus the number of transits is shown in Fig. 12. For each of the two sections, the figure provides the model outcome for the various scenarios (Table 6) and the two vessels (the cargo and the cruise ship) addressed by the simulations. The largest mass seepage occurs at the beginning, then decreases due to the $c$ reduction in the volume surrounding the channel bottom and slopes (Fig. 13). Scenario-1 does not differentiate appreciably from the Base scenario, suggesting a negligible effect of the transversal dispersivity. Notice that, although the maximum groundwater velocities were computed in Section-1, $m^*$ is larger (from 1.9 to 3.5 time) in Section-2 independently on the dispersivity scenario because of the larger subsurface volume with significant velocity values in this latter (Figs. 11c and 11d). Table 7 summarizes the ratio between the expelled mass at $N_v = 1000$ as obtained with Scenario1-Scenario3 and the reference values provided by the Base case. Decreasing and increasing $\alpha_L$ of one order of magnitude yields a reduction and rise of $m^*$ in the range of 20-40% and 220-370%, respectively.

Combining the actual initial concentration of the various anthropogenic contaminants (Table 5) and the TRAN3D outcomes allows estimating the real mass m of each contaminants released into the lagoon. Fig. 14 shows the behaviour of m versus the number of ship transits. The profiles have been obtained by averaging the outcomes obtained for Section-1 and Section-2 and using a MVC length equal to 3 km.

### 4.3 Aquifer salinization

Fig. 15 shows the outcome of COUPHYB in terms of relative concentration at the end of the simulation period, i.e. 10 years after the inception. The results are presented for both the sections addressed by the study. The effect of the MVC excavation is pointed out by comparing the two setting, i.e. the present condition and that where the MVC is dredged. Cutting the top clayey layer, the excavation favours the propagation at depth of the seawater, with an increase of $c$ from the initial 0.7 value to more than 0.9 in the surrounding of the canal. The salt propagation downward is more pronounced in Section-2 than in Section-1 because of the Aqt-2 presence directly below the MVC bottom in the latter. Notice that the increased contamination remains located in the surrounding of the channel excavation, with Aqt-3 completely precluding the salt transport at larger depth.

### 4.4 Model evaluation

We are aware that the analyses presented here rely on a number of simplifications, with the results that can be affected by (1) the approximated modelling approach use for the simulations, (2) the representativeness of the hydrologic and geological information used to calibrate the model parameters, and (3) the boundary conditions and the factors forcing the system (Tsang, 2005).



In this study, we have elected to use a 2D modelling approach along vertical sections instead of a 3D analysis. The choice is warranted by the shape of the domain possibly influenced by the excavation, which is much more elongated along the MVC trace than in the orthogonal direction. Moreover, the groundwater flow induced by a ship moving along the canal, which

represents a main factor forcing the system in the short and medium term, is characterized by a net component along the direction orthogonal to the ship track only. The two selected sections are representative of the main variability characterizing the hydrogeological architecture of the Venice Lagoon subsurface. Several boreholes drilled in the past and the extensive seismic survey carried out during the initial phase of the study provided an accurate characterization of the shallowest 50-m thick depositional sequence and revealed that, although local sedimentary anomalies are frequently encountered, the

presence of a buried large channelling system cutting Aqt-2 in a specific portion of the study area is the main feature to be accounted in the modelling investigations.

A critical issue is related the calibration of the hydrogeological parameters ($K$, $S_s$, $D$). The model calibration has been conducted using trial-and-adjustment (Anderson et al., 2015) and relies on the piezometric records collected at two boreholes, one per section, penetrating the MVC depth of excavation. We are aware of the non-uniqueness of this solution

and the paucity of the data that do not support, for example, to account for a possible spatial variation of the parameters. Concerning $K$, isotropic porous medium has always been assumed in the simulations described above. Two reasons support the choice of neglecting anisotropy: firstly, the shallow depth of interest that reduces the effect of the geostatic load in decreasing $K$ along the vertical direction (Whipkey and Kirkby, 1978); secondly, bioturbation that enhances the "original" hydraulic conductivity of sedimentary units along the vertical direction. Such activities are typically limited to less than a

meter in depth (Gerino et al., 2007), but can affect large thickness in highly dynamic coastal depositional environments (Gingras et al., 2015). However, notice that a precise calibration of the model is beyond the scope of the study. Indeed, the principal goal is to investigate which are the main processes affecting a lagoon subsurface caused by a deep excavation and transit of large ships through a shallow tidal flat.

Concerning the transport model, a sensitivity analysis on $\alpha_L$ and $\alpha_T$ has been carried out. Due to the lack of specific

dispersion and tracer tests, the amount of contaminants released from the shallow sediments into the lagoon has been quantify versus the $\alpha_L$ and $\alpha_T$ values. In this first analysis, the transport simulations have been focussed only on the labile fraction of the contaminants, thus supporting the use of a non-reactive transport model. The spatial and depth variability of the initial contaminant concentrations has been properly characterized in eleven boreholes, continuously cored along the MVC trace (Figure 1).

Finally, the simulations have been carried out assuming a null natural flow due to the lack of specific information. Because of the shallow depth of investigation and the position of the study area at the edge of the flat Po Plain, the natural hydraulic gradient is certainly very small. Therefore, the results of the short- and mid-term simulations are almost unaffected by the assumption.





## 5 Conclusive discussion

Lagoons are natural environments that undergo a continuous increase of anthropogenic pressure. On the other hand, their present hydraulic and morphologic equilibrium is in some cases artificially preserved by human interventions, which are aimed at contrasting the combined effects of sea level rise associated to global warming and land subsidence of natural and anthropogenic origin.

The Lagoon of Venice is a paradigm of the complexity in the interactions among economic, social, and environmental needs (Rinaldo, 2001). This holds both for the surface and for subsurface environments. Investigations carried out over the last years (Teatini et al., 2011; Viezzoli et al., 2010) revealed that fresh groundwater resources are found at quite low depths, i.e., 30–40 m and even less than 10 m beneath the lagoon bottom. Like the hydrogeological setting of other lagoons (e.g., Santos et al., 2008), the silty-clayey layer marking the boundary between the marine Holocene and continental Pleistocene deposits precludes or at least reduce the vertical leakage of the salt waters downward into the underlying fresh-water aquifers. However, a large petrochemical industrial district, the PMIZ, has been in operation since the 1950s at the lagoon-mainland interface representing a main source of soil and water pollution around the area (e.g., Zonta et al., 2007). Despite an almost 50-km long cut-off wall built-up along the canal banks of the PMIZ to prevent discharge of contaminated waters into the lagoon (Paris et al., 2011), results from chemical analyses provided evidence of a high content of Hg, Zn, and other metals in the bottom sediments and pore water not only in front of the industrial site (Gieskes et al., 2015) but also at distance. Although quite gentle in the shallower subsurface, the natural groundwater flow from the mainland seaward has likely transported the contaminants to the lagoon ecosystem over the last decades.

The quality of an aquatic ecosystem is set by the quality of its sediments. Sediments are a sink for pollutants and nutrients, but can also act as a long-term source as well, with the groundwater playing a key role in the redistribution of hazardous substances in other environmental compartments, such as the biota, upon changes in the physic-chemical conditions.

The evaluation of the possible impacts of the MVC excavation must be investigated in this context. Cutting of the clayey layers that characterize the shallower Pleistocene and Holocene deposits in coastal zones can significantly increases the exchange between groundwaters and surficial water bodies and the anthropogenic and/or natural contaminants transported with the waters. For example in the Mangueira Lagoon, which is a large (90 km long), shallow (~4–5 m deep), fresh, and non-tidal coastal lagoon in southern Brazil surrounded by extensively irrigated rice plantations and numerous irrigation channels, the use of naturally geochemical tracers ($^{222}$Rn, $^{223}$Ra, $\delta^{18}$O, $\delta^{2}$H and others) showed that dredging of irrigation channels altered the SGD fluxes (Santos et al., 2008). In spite of the relatively small depth and dimension of the channels, the ditch digging cut the shallow aquitards, which previously restricted the upward advection from the permeable strata underlying the lagoon bottom, increasing the fluxes of contaminants into the lagoon.

The modelling study presented in this work provides a first evaluation of how the MVC can affect the subsurface lagoon environment. The interruption of the caranto aquitard favours the saltwater flow deepward in a medium to long time interval, in the range of a few decades. Electromagnetic surveys and marine electric topographies carried out in the part of the lagoon




between Venice and Chioggia clearly pointed out that groundwater with a salt content similar to the marine waters is found beneath 5-15 m below msl only where the caranto layer is cut, generally by natural erosion or channel excavation (Tosi et al., 2009; Zecchin et al., 2014). The modelling results suggest that the salt contamination remain localized around the incision, with an important role in controlling the depth of percolation played by the actual layering of the sedimentary deposits below

the channel bottom.

A significant influence on the groundwater – surficial water exchange is expected to be produced by the transit of cruise vessels. Each large ship in transit along the channel can produce a depression wake of the order of 1 m, thus pumping out the groundwater from the shallow deposits around the excavation. Although for a given channel section the ship-wake lasts a couple of minutes only, the large groundwater velocity induced in the surroundings of the excavation combined with the

length of the MVC are responsible for an efflux in the order of 50-100 $m^3$ per ship, i.e. 25'000-50'000 $m^3$/year. Similarly to other coastal lagoons with an inner port and/or an industrial zone, for example the Maryut Lagoon, Nile Delta (Oczkowsly and Nixon, 2010), or the Lake Macquarie, New South Wales, Australia (Thomsen et al., 2009), anthropogenic contaminants have been detected in the lagoon subsurface. The contaminants in the labile and bioaccessible fraction along the MVC designed path and depth range might be released into the lagoon because of ship-wakes, with a considerable amount in the

mid-term. The potential bioavailability and bioaccessibility of contaminants was confirmed by the high ecotoxicity levels shown by elutriates obtained from sediment samples collected along the MVC trace. They generate concern as it represents an easily exchangeable fraction which can move from sediment to water. Proper measures should be then planned to limit the risk of contamination of the lagoon water during the years following dredging. Moreover, considering the importance and the fragility of the Lagoon of Venice, if the MCV will become a real project, a number of new and more detailed

information will be necessarily collected to provide a more accurate quantification of the possible environmental impacts of the canal dredging on the subsurface system of the Venice Lagoon. For example, additional piezometers should be placed along the MMIC to verify in advance the ship-wake effects on the subsoil and along the planned MVC path to characterize the natural flow regime; groundwater age through isotope analyses on water samples should be determined to evaluate the groundwater origin and fate; pumping and tracer tests should be planned to characterize the hydrogeological properties of the

shallow aquifers below the lagoon bottom.

*Data availability*. All data are available upon request to Pietro Teatini (pietro.teatini@unipd.it).

*Competing interests*. The authors declare that they have no conflict of interest.

*Acknowledgements*. The research was funded by the Venice Port Authority, Italy, and partially supported by the Flagship Project RITMARE - The Italian Research for the Sea, CNR-MIUR, National Research Program 2011-2013, "Linea SOLVE".



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



## Tables

**Table 1. Total concentration, labile fraction, and percentage of labile fraction (calculated with respect to the total concentration) of Cr versus depth in the Section-1 borehole. The ratio dry weight/wet weight of the sediments ranges between 0.17 and 0.37.**

| Depth below the lagoon bottom [m] | Cr - total [mg/Kg dry weight] | Cr - labile fraction [mg/Kg wet weight] | Cr % labile |
|---|---|---|---|
| 0-1 | 100.8 | 0.32 | 1 |
| 1-2 | 130.8 | 0.26 | 0 |
| 2-3 | 53.4 | 0.70 | 2 |
| 3-4 | 40.3 | 0.51 | 2 |
| 4-5 | 20.9 | 0.45 | 6 |
| 5-6 | 23.4 | 0.65 | 6 |
| 6-7 | 27.8 | 1.33 | 28 |
| 7-8 | 25.5 | 1.16 | 25 |
| 8-9 | 15.7 | 1.47 | 63 |
| 9-10 | 18.4 | 1.75 | 72 |

**Table 2. Geometry of the typical cruise vessel used in the uRaNSe-Xnavis and SHYFEM simulations.**

| Lenght [m] | 300 |
|---|---|
| Width [m] | 32 |
| Depth [m] | 8.7 |
| Blocking coefficient [–] | 0.7 |

**Table 3. Section-1: computed maximum depression (m) in the MVC and percentage attenuation at various distance from the**
10 **channel center for different speed of the cruise ship. Similar values are obtained in Section-2.**

| Distance from channel [m] | 3.1 knots | 4.9 knots | 5.9 knots | 7.7 knots |
|---|---|---|---|---|
| 0 | 0.15 m | 0.3 m | 0.4 m | 0.9 m |
| 30 | -36.4 % | -19.1 % | -19.2 % | -16.5 % |
| 88 | -40.6 % | -17.8 % | -19.7 % | -12.5 % |
| 177 | -62.1 % | -44.5 % | -38.5 % | -31.8 % |
| 358 | -68.5 % | -64.5 % | -54.0 % | -48.9 % |
| 594 | -78.8 % | -74.6 % | -69.7 % | -64.1 % |





**Table 4. Hydrogeological parameters obtained by the model calibration and used in the numerical simulations. Aqt-2 lacks in Section-2.**

| Layer | $K$ [m/s] | $S_s$ [m$^{-1}$] |
|---|---|---|
| Aqt-1 | $1.7\times10^{-6}$ | $5.0\times10^{-4}$ |
| Aqf-1 | $1.0\times10^{-5}$ | $2.0\times10^{-5}$ |
| Aqt-2 | $1.0\times10^{-7}$ | $5.0\times10^{-4}$ |
| Aqf-2 | $1.0\times10^{-5}$ | $2.0\times10^{-5}$ |
| Aqt-3 | $1.0\times10^{-7}$ | $5.0\times10^{-4}$ |
| Aqf-3 | $1.0\times10^{-5}$ | $2.0\times10^{-5}$ |
| Aqt-4 | $1.0\times10^{-7}$ | $5.0\times10^{-4}$ |

**Table 5. Average contaminant concentration (labile fraction) in the depth range between -9 and -5 m below msl on Section-1 and Section-2.**

| Contaminant | $c$ [mg/l] Section-1 | $c$ [mg/l] Section-2 |
|---|---|---|
| Hg | 0.03 | 0.01 |
| Cd | 0.08 | 0.04 |
| Pb | 0.99 | 0.30 |
| As | 0.47 | 0.17 |
| Cr | 0.88 | 0.68 |
| Cu | 0.59 | 0.14 |
| Ni | 0.35 | 0.72 |
| Zn | 1.50 | 1.03 |
| Se | 0.40 | 0.04 |
| V | 0.67 | 0.90 |
| Sb | 0.02 | 0.01 |

**Table 6. Longitudinal ($\alpha_L$) and transversal ($\alpha_T$) dispersivity for the scenarios investigated with TRAN3D.**

| Scenario | $\alpha_L$ [m] | $\alpha_T$ [m] |
|---|---|---|
| Base | 1 | 0.1 |
| Scenario1 | 1 | 1 |
| Scenario2 | 10 | 1 |
| Scenario3 | 0.1 | 0.1 |





**Table 7. Ratio between the reference mass *m*\* expelled from the subsurface into the MVC after a transit of 1000 ships for the various scenarios addressed by the study.**

| Scenario | Ship | Section | |
|---|---|---|---|
| | | **Section-1** | **Section-2** |
| Scenario-1/Base | Cargo | 1.01 | 1.01 |
| | Cruise | 1.01 | 1.00 |
| Scenario-2/Base | Cargo | 2.28 | 3.35 |
| | Cruise | 2.34 | 3.70 |
| Scenario-3/Base | Cargo | 0.39 | 0.22 |
| | Cruise | 0.35 | 0.18 |



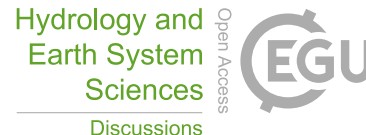

**Figures**

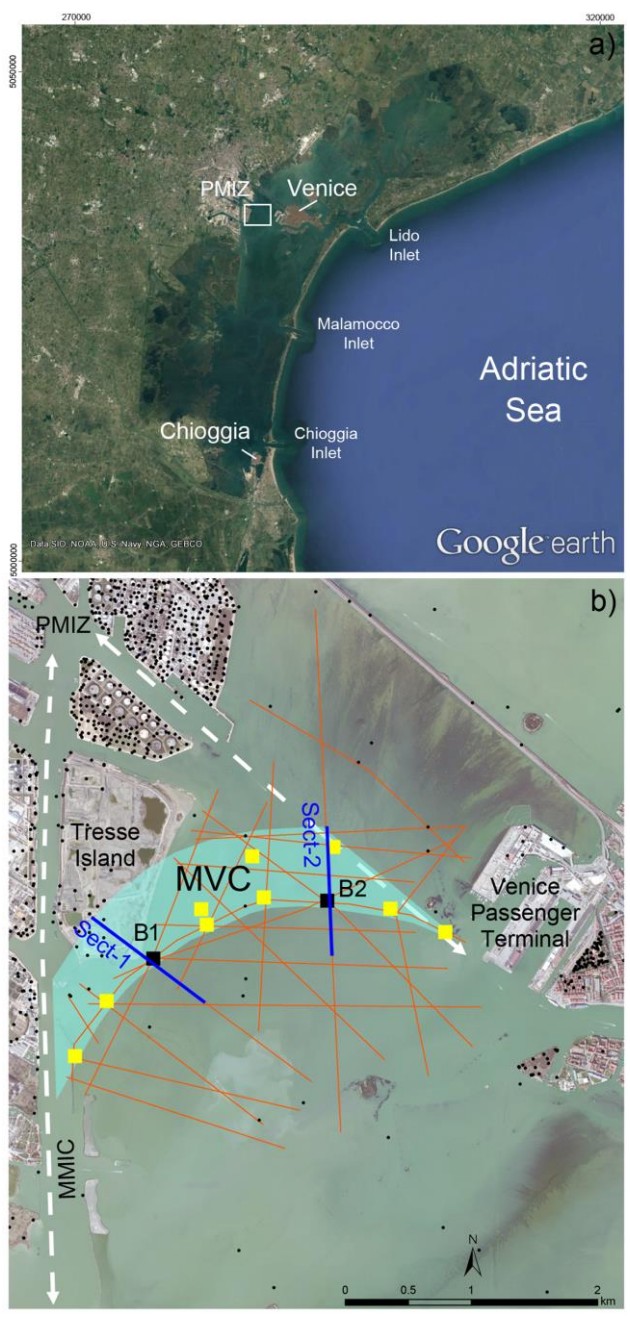

**Figure 1: (a) Satellite view of the Venice Lagoon and (b) map of the study area. The zone involved by the MVC dredging is highlighted in light blue with the main navigation canals shown by the dashed with alignments. The traces of Section-1 and Section-2 are shown in blue and those of the seismic survey in red. The square dots represent the location of the new 10-m deep boreholes; the small black dots indicate the positions of hydro-stratigraphic previous investigations.**



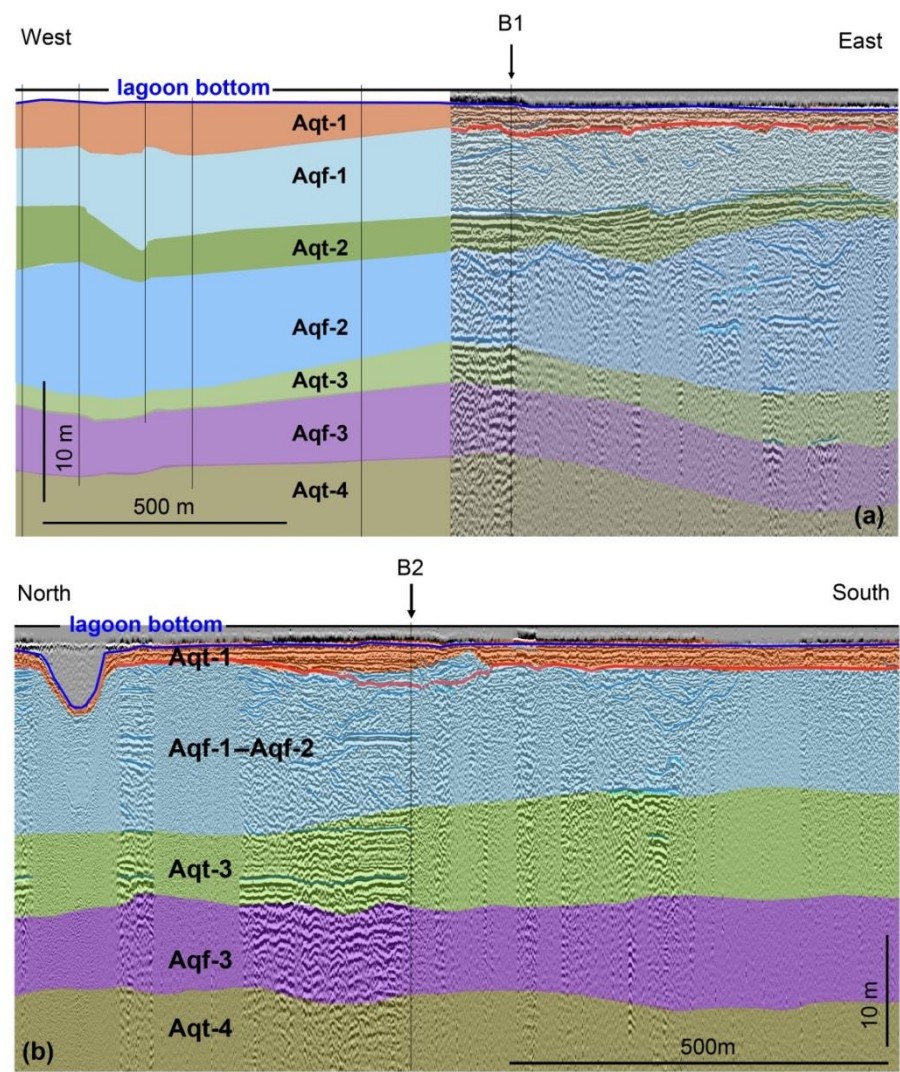

**Figure 2: Hydrogeological setting of the subsurface of the Venice Lagoon along (a) Section-1 and (b) Section-2 (see Figure 1 for their location) as obtained by the interpretation of the seismic acquisitions integrated with wellbore lithostratigraphies. The borehole locations are shown by B1 and B2.**





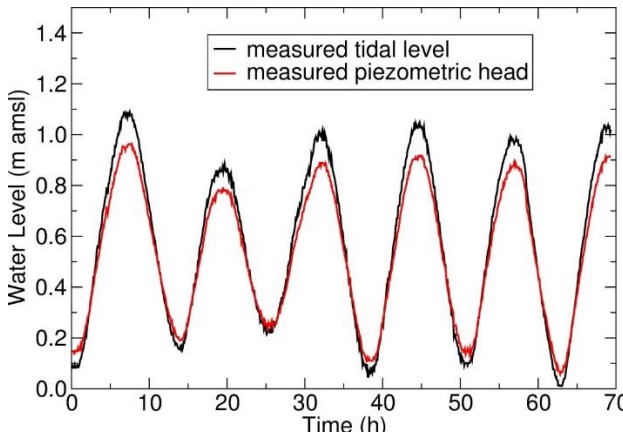

**Figure 3: Measured tidal level and piezometric head at the Section-2 borehole over the period between March 7 and March 10, 2016.**



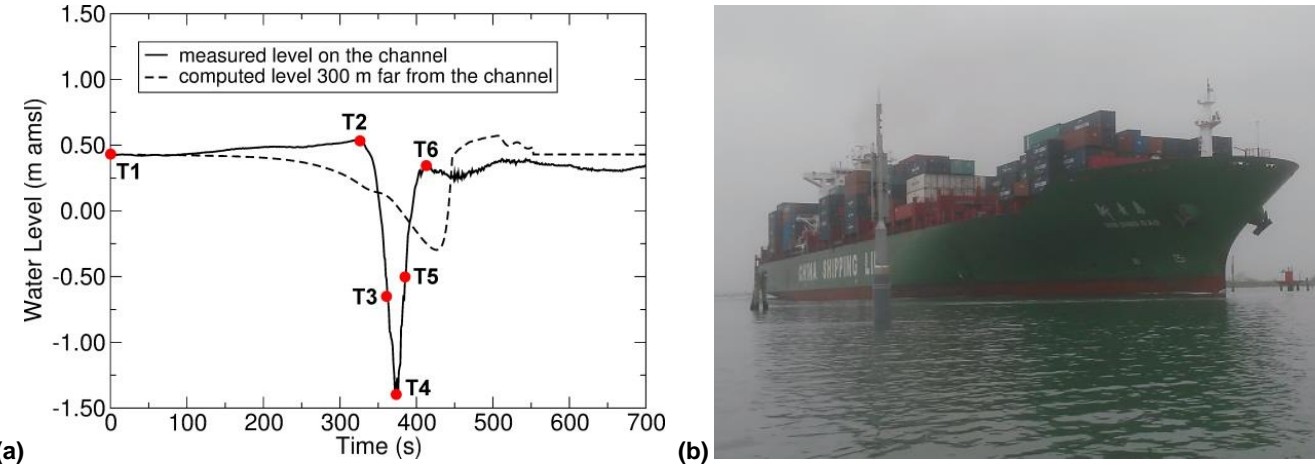

(a)                                                                  (b)

**Figure 4: (a) Water level (m above msl) recorded on the bottom of the MMIC and computed by SHYFEM on the tidal flat 300 m far from the channel caused by the transit on April 6, 2016, of the Cargo-Hazard A. A photo of the vessel is shown in (b). The commercial vessel, which was used as reference, is 280 m long, 40 m wide, and characterized by a gross tonnage of 66433 t. The ship speed $s$ was equal to 8.1 knots (4.2 m/s). The time steps T1-T6 highlighted in (a) refer to the model outcome shown in Fig. 10.**





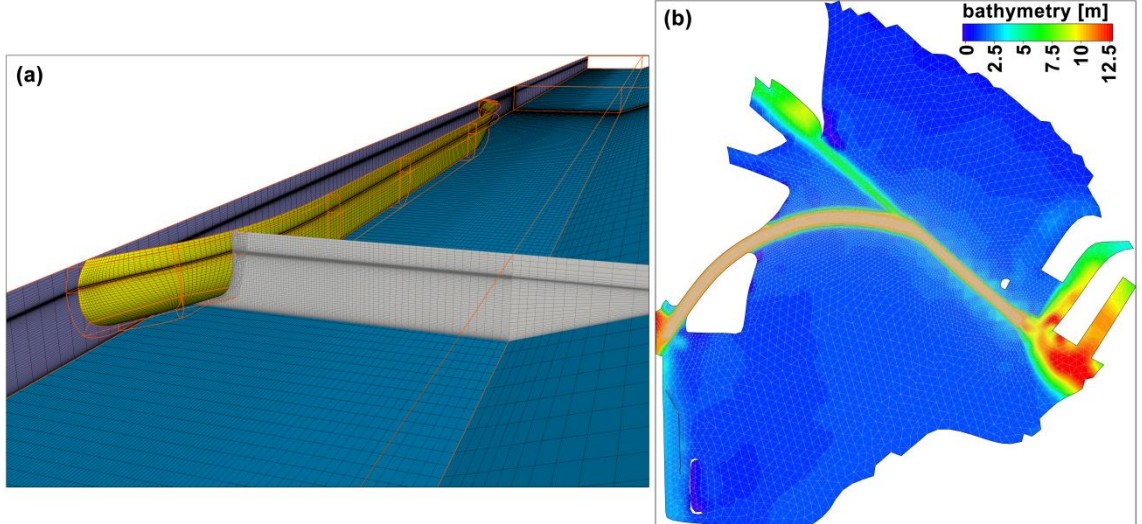

**Figure 5: (a) Computational grid adopted in the uRaNSe-Xnavis simulations along the MCV. The figure shows the (coarse) grid over the ship hull (yellow), with the longitudinal symmetry plane shown in purple and the channel bottom in blue. (b) Finite element grid used in SHYFEM to represent the bathymetry of the Venice Lagoon portion where the MVC is expected to be dredged.**





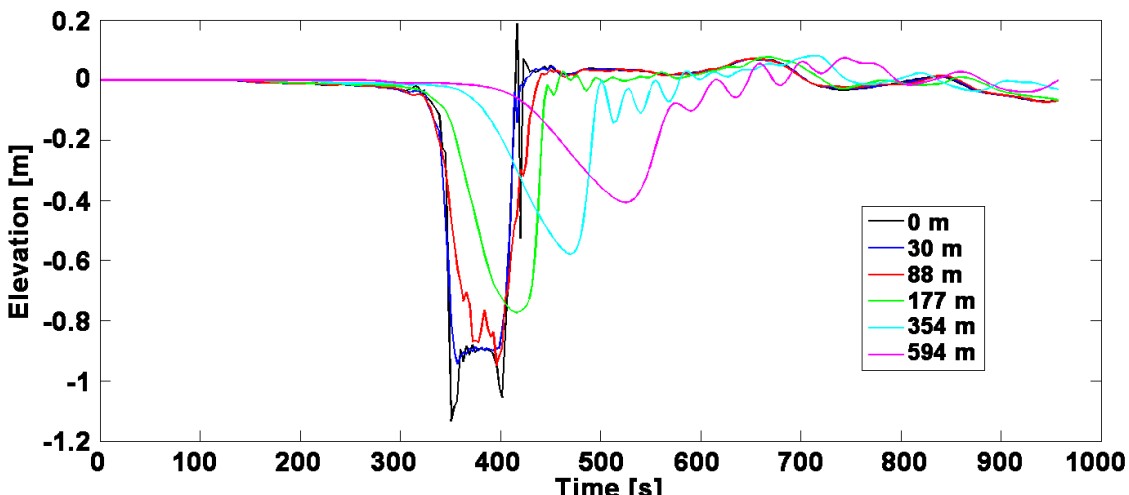

**Figure 6: Water level behaviour at various distance from the centre of the MVC as computed by the hydrodynamic model for a liner ship moving at *s* = 7.7 knots.**



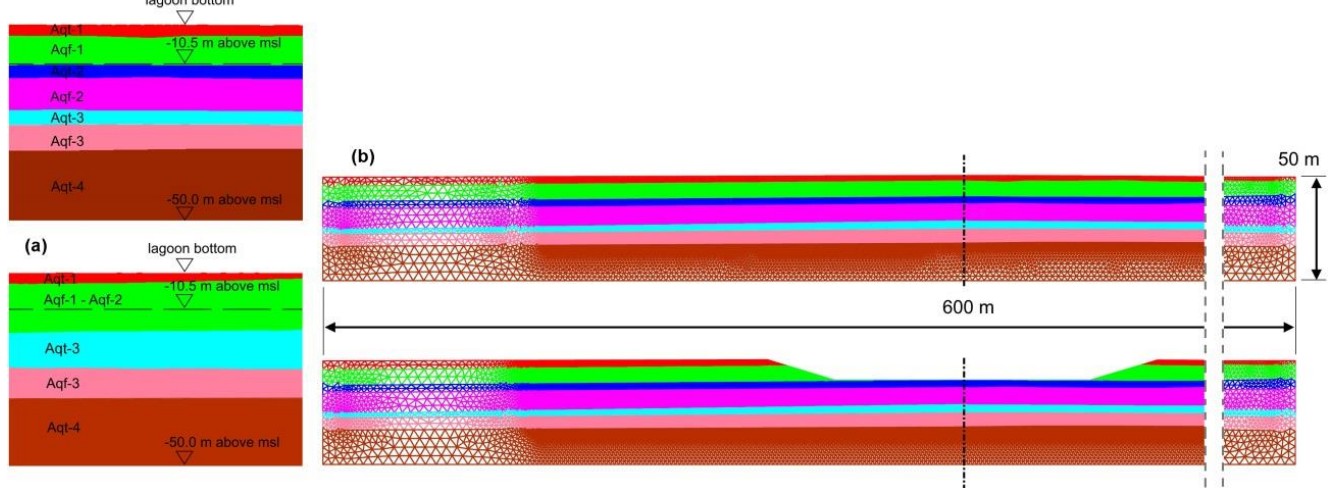

**Figure 7: (a) Hydrogeological setting of Section-1 (above) and Section-2 (below) with respect to the MVC bottom. (b) Finite element mesh of Section-1 prior (above) and after (below) the MVC excavation. The colours are representative of the various soil types.**




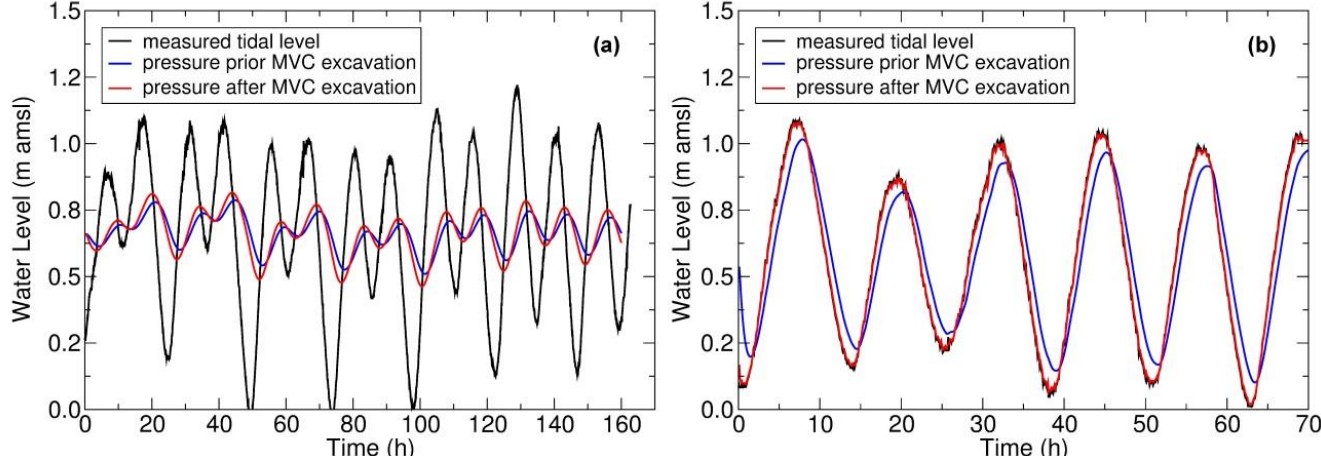

**Figure 8: Behaviour versus time of the pressure at a depth of 13 m below msl in (a) Section-1 and (b) Section-2, respectively, prior and after the MVC excavation. The tide fluctuation is provided for comparison.**





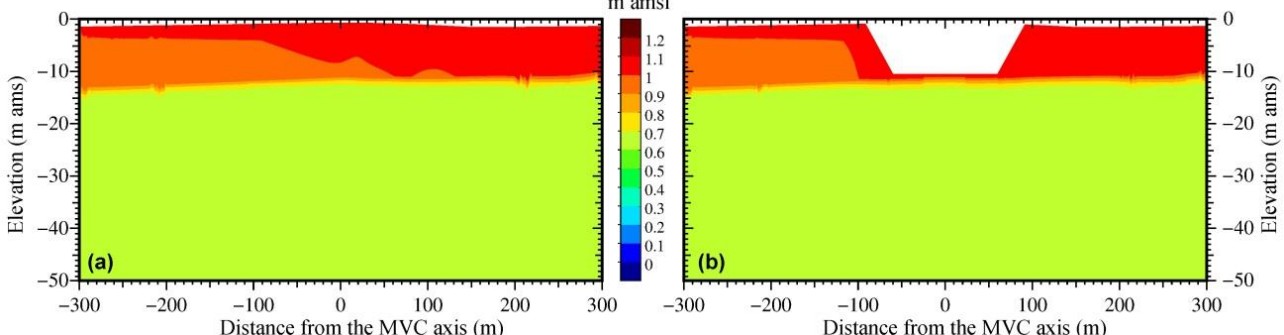

**Figure 9: Section-1: computed pressure distribution (a) prior and (b) after the MVC excavation at the maximum tidal level highlighted in Fig. 8. Vertical exaggeration is 8.**




**Figure 10: Section-2: computed pressure distribution at the times T1-T6 highlighted in Fig. 4a during the transit of the cargo liner of Fig. 4b. The vertical exaggeration is 8.**





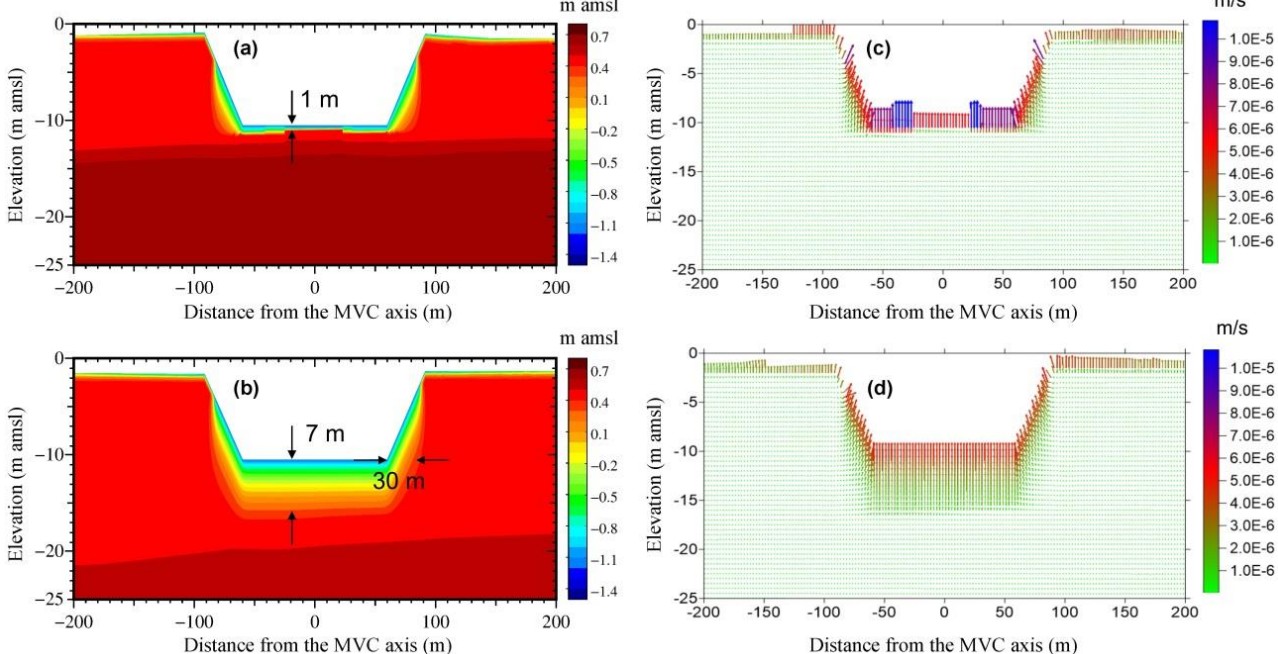

**Figure 11: Computed (a,b) pressure distribution and (c,d) velocity field at the maximum wake induced by the ship transit in (a,c) Section-1 and (b,d) Section-2. The dimensions of the subsurface portion feeling the ship transit are highlighted in (a,b). Vertical exaggeration is 8.**




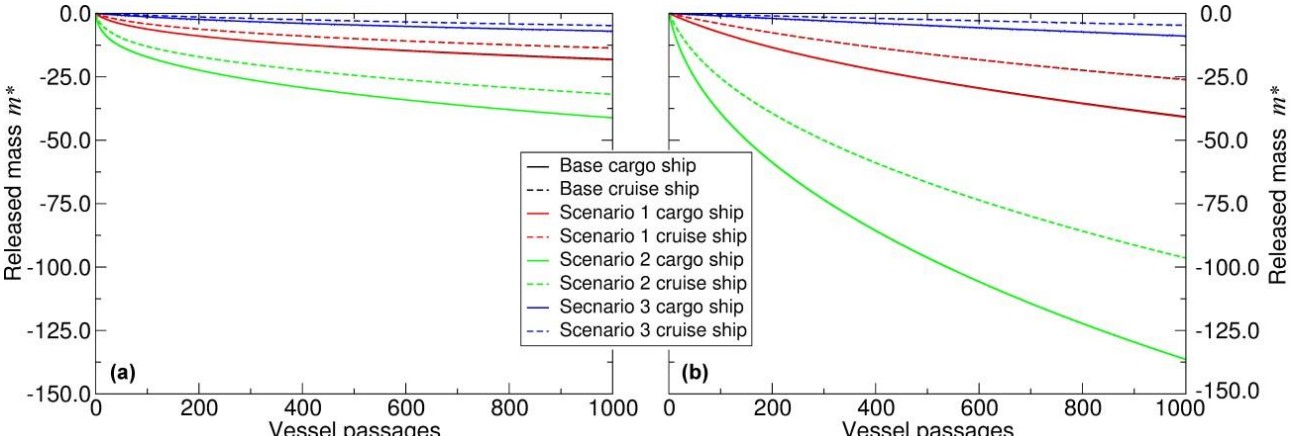

**Figure 12: Behaviour of *m\** as computed by TRAN3D in (a) Section-1 and (b) Section-2, respectively, for the different dispersivity scenarios and the two ship types addressed by the simulations.**




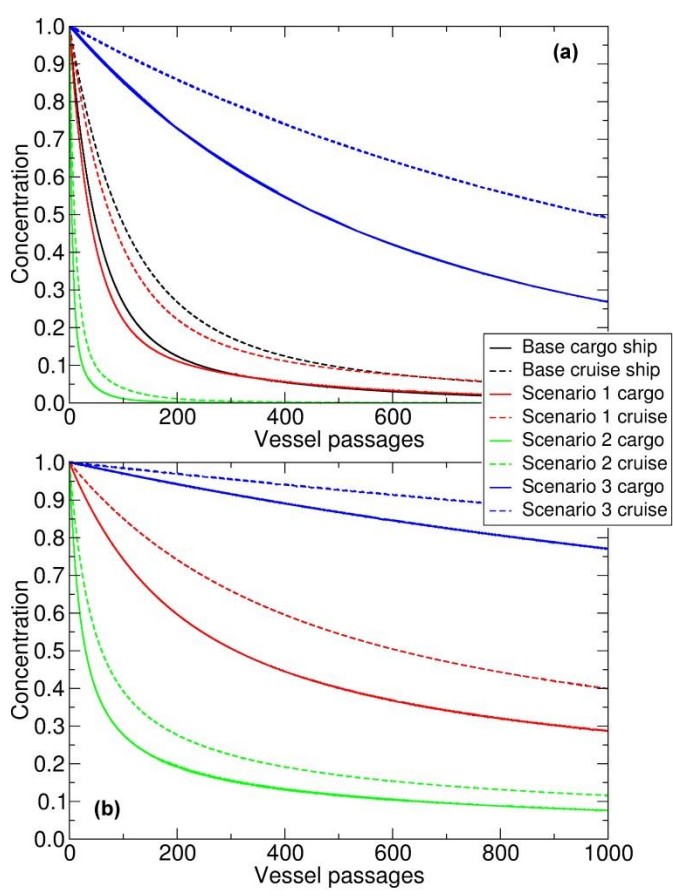

**Figure 13: Section-2: Behaviour of *c* for a node located 0.5 m below the MVC bottom along the symmetry axis as computed by TRAN3D in (a) Section-1 and (b) Section-2, respectively, for the different dispersivity scenarios and the two ship types addressed by the simulations.**




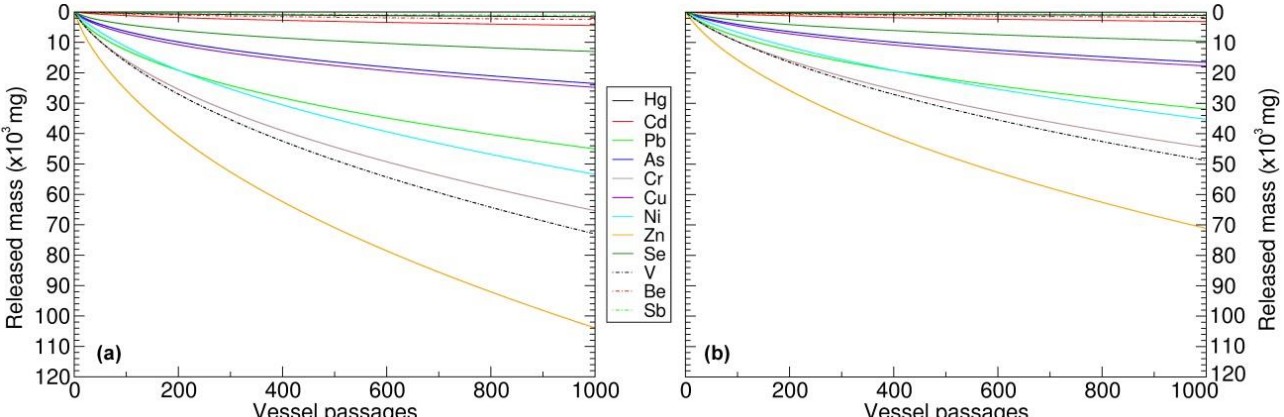

**Figure 14: Mass of various chemicals expelled from the lagoon subsurface through the MVC bottom versus the transit number of (a) cargo and (b) cruise ships.**



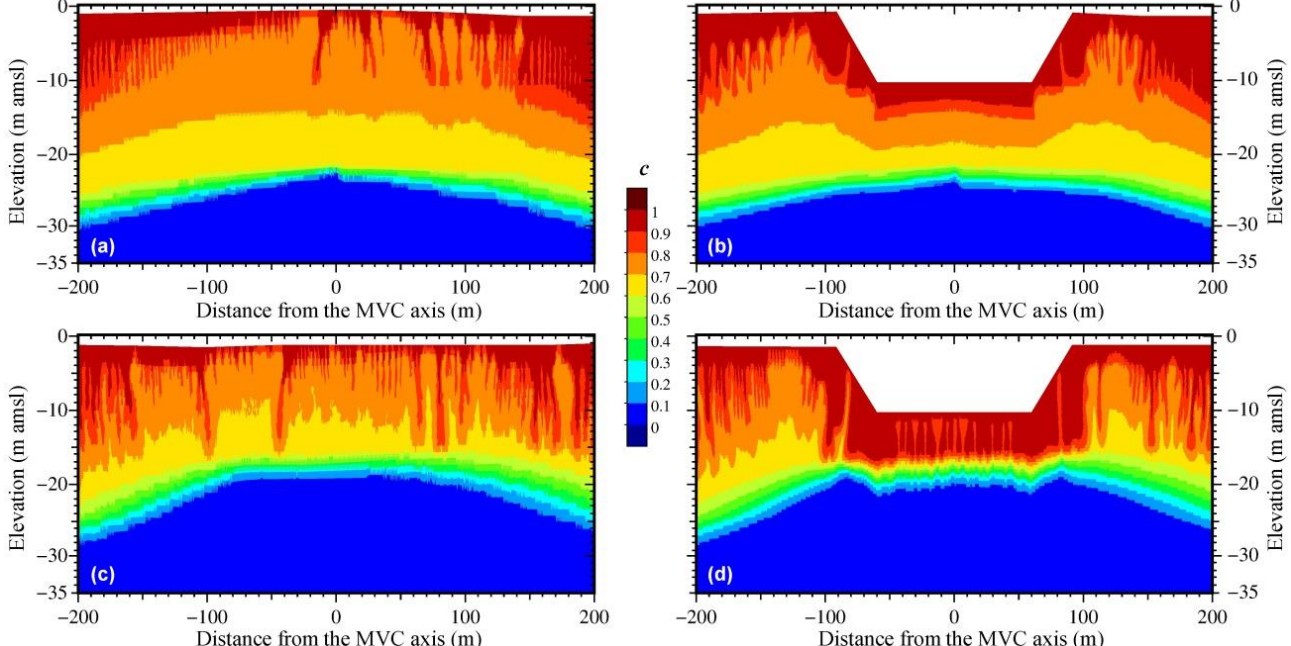

**Figure 15: Relative salt concentration in (a,c) initial conditions and (b,d) after 10 years as computed in (a,b) Section-1 and (c,d) Section-2 in the present (a,c) and in the planned (b,d) conditions, respectively. Vertical exaggeration is 12.**