# Peer review of "Hydrogeological effects of dredging navigable canals through lagoon shallows. A case study in Venice"

_Hydrology and Earth System Sciences, 2017_

## Referee Comment (RC1) · Anonymous Referee #1 · 28 Jun 2017

The manuscript is very interesting and has a large amount of data (hydrological, geochemical, geophysical) that are integrated with a modeling. I suggest some modifications that will help to better understand the analysis of data and to make reading more enjoyable. - The manuscript is very extensive and its reading is complicated in the current format. I suggest clearly indicating the methodology, results, discussion and conclusion in separate items. - Add a more detailed description of the chemical results. - I understand that labile oligoelement is that fraction of sediments that is readily transformed into soluble. But, is really soluble when the ship transits on the channel? Justify the desorption of these elements. - I suggest inserting the sediment chemistry data into the profiles of Figure 2 and deleting Tables 1 and 5 - Replace mg/l per mg/L -

I do not specialize in models, I suggest that another reviewer see these items.

---

## Referee Comment (RC2) · Anonymous Referee #2 · 7 Jul 2017

This paper is absolutely interesting for the evaluation on the hydogeological effects brought by intensified human activities in the coastal zones. This paper presents a large amount of data observed or modeled. I agree on the comments from another reviewer. These data can be shown in a better way for a enjoyable reading. Table 5, I think is not necessary to show so many elements. You can show several typical elements and the potential changes due to the dredging. Fig.1, " the small black dots indicate the positions of hydro-stratigraphic previous investigations." If no data is shown with these points, they should be removed for a better illustration. Fig.4, a description on the vessel is necessary, but not with the photo. what are the red points? why not also show the computed results together with the measured one since you can

compute that 300 m far away?

I don't think 4.4 model should be in the section of "results". A section of discussion may be better.

––––––––––––––––––––––––––––

---

## Author Comment (AC1) · 27 Jul 2017

We are grateful to Reviewer #2 for the generally positive evaluation of our work. To improve the quality of this manuscript, we carefully revised the text by incorporating the comments one by one. The detailed revisions are presented in the responses to each comment.

Comment 1: The manuscript is very extensive and its reading is complicated in the current format. I suggest clearly indicating the methodology, results, discussion and conclusion in separate items.

[Figure]

Response: The manuscript has been updated following this reviewer suggestion. The new structure comprises the following sections: available data, modelling, results, discussion and conclusion.

Comment 2: Add a more detailed description of the chemical results. I understand that labile oligoelement is that fraction of sediments that is readily transformed into soluble. But, is really soluble when the ship transits on the channel? Justify the desorption of these elements.

Response: Section 2.3 has been integrated with a more detailed description of the chemical characterization. It has been specified better that the labile fraction is readily exchangeable and therefore can be immediately moved as soon as the pressure wave generated by a ship transit increases the groundwater flow toward the channel. Two new references have been added to help keep the section with a length comparable with those of the other issues.

Comment 3: I suggest inserting the sediment chemistry data into the profiles of Figure 2 and deleting Tables 1 and 5.

Response: Table 1 has been removed with the profiles added in Figure 2 (new insets, Figure 2c). Conversely, we prefer to keep Table 5 for clearness.

Comment 4: Replace mg/l per mg/L.

Response: Done.
* * *

---

## Author Comment (AC2) · 27 Jul 2017

We are grateful to Reviewer #2 for the positive evaluation of our work. To improve the quality of this manuscript, we carefully revised the text by incorporating the comments one by one. The detailed revisions are presented in the responses to each comment.

Comment 1: Table 5, I think is not necessary to show so many elements. You can show several typical elements and the potential changes due to the dredging.

Response: The reason why we decided to analyze and present the concentration of 12 ions is because these are the elements for which the current EU law in the field of

water policy (directive 2013/39/eu) requires a specific analysis. This has been explicitly reported in section 2.3. For this reason we think that it is preferable to keep all the information in the paper.

Comment 2: Fig.1, "the small black dots indicate the positions of hydro-stratigraphic previous investigations." If no data is shown with these points, they should be removed for a better illustration.

Response: We agree with the reviewer suggestion. We have removed the majority of the "black dots" and kept only those located in the surroundings of the seismic survey and used to drive the inversion of the geophysical acquisitions. The position of a few of them are also reported in Fig. 2a (black thin lines).

Comment 3: Fig.4, a description on the vessel is necessary, but not with the photo. what are the red points? why not also show the computed results together with the measured one since you can.

Response: the photo has been removed. The red dots represent the times (level conditions) at which the model outcomes are presented. The figure caption has been updated to make it clearer. We prefer to show only the level profiles directly used as boundary conditions in the hydrogeological models (i.e. the measurements in the channel and the result of the hydrodynamic modeling approach in the tidal flat). The paper is not focused on hydrodynamic modeling and an example of how the ship-wakes evolve in space is already provided in Figure 6.

Comment 4: I don't think 4.4 model should be in the section of "results". A section of discussion may be better.

Response: Done. The section has been moved to the new Discussion section

The manuscript revised according with recommendations by Referees #1 and #2 is provided in the supplement.

Please also note the supplement to this comment:
https://www.hydrol-earth-syst-sci-discuss.net/hess-2017-317/hess-2017-317-AC2-
supplement.pdf

———————————————————
317, 2017.

[Figure]

**Supplement:**

[revised manuscript text omitted]

---

## Author Comment (AC3) · 27 Jul 2017

Here the new Figure 1 updated as suggested by Referee #2

[Figure]

[Figure]

**Fig. 1.**

---

## Referee Comment (RC3) · Anonymous Referee #1 · 28 Jul 2017

The manuscript has been modified considering the suggestions of all the reviewers. I suggest can be accepted in the current format
* * *